# Comprehensive Transcriptional Profiling and Mouse Phenotyping Reveals Dispensable Role for Adipose Tissue Selective Long Noncoding RNA *Gm15551*

**DOI:** 10.3390/ncrna8030032

**Published:** 2022-05-06

**Authors:** Christoph Andreas Engelhard, Chien Huang, Sajjad Khani, Petr Kasparek, Jan Prochazka, Jan Rozman, David Pajuelo Reguera, Radislav Sedlacek, Jan-Wilhelm Kornfeld

**Affiliations:** 1Department for Biochemistry and Molecular Biology (BMB), University of Southern Denmark, Campusvej 55, 5230 Odense, Denmark; christophak@bmb.sdu.dk (C.A.E.); f04626009@ntu.edu.tw (C.H.); 2Laboratory of Animal Physiology, Department of Animal Science and Technology, National Taiwan University, Taipei 10617, Taiwan; 3Max Planck Institute for Metabolism Research, Gleueler Strasse 50, 50931 Köln, Germany; khani.sajjad@gmail.com; 4Institute for Diabetes and Cancer (IDC), Helmholtz Zentrum München, German Research Center for Environmental Health, 85764 Neuherberg, Germany; 5Cluster of Excellence Cellular Stress Responses in Aging-Associated Diseases (CECAD), Faculty of Medicine, University Hospital of Cologne, University of Cologne, Joseph-Stelzmann-Str. 26, 50931 Cologne, Germany; 6Czech Centre for Phenogenomics, Institute of Molecular Genetics of the Czech Academy of Sciences, Prumyslova 595, 25250 Vestec, Czech Republic; petr.kasparek@img.cas.cz (P.K.); jan.prochazka@img.cas.cz (J.P.); jan.rozman@img.cas.cz (J.R.); david.pajuelo-reguera@img.cas.cz (D.P.R.); radislav.sedlacek@img.cas.cz (R.S.)

**Keywords:** long noncoding RNAs, brown adipose tissue, adipose tissue remodelling

## Abstract

Cold and nutrient-activated brown adipose tissue (BAT) is capable of increasing systemic energy expenditure via the uncoupled respiration and secretion of endocrine factors, thereby protecting mice against diet-induced obesity and improving insulin response and glucose tolerance in men. Long non-coding RNAs (lncRNAs) have recently been identified as fine-tuning regulators of cellular function. While certain lncRNAs have been functionally characterised in adipose tissue, their overall contribution in the activation of BAT remains elusive. We identified lncRNAs correlating to interscapular brown adipose tissue (iBAT) function in a high fat diet (HFD) and cold stressed mice. We focused on *Gm15551*, which has an adipose tissue specific expression profile, is highly upregulated during adipogenesis, and downregulated by β-adrenergic activation in mature adipocytes. Although we performed comprehensive transcriptional and adipocyte physiology profiling in vitro and in vivo, we could not detect an effect of gain or loss of function of *Gm15551*.

## 1. Introduction

The prevalence of obesity is increasing worldwide [1]. Obesity is the result of a chronic imbalance between energy intake and expenditure, resulting in the accumulation of excess adipose tissue. Obesity is correlated with increased overall mortality and is a risk factor for various diseases, including cardiovascular disease and diabetes type 2 [2,3].

Adipose tissue plays a central role in the regulation of energy balance. While white adipose tissue (WAT) mainly functions as the storage of excess energy in the form of triglycerides, brown adipose tissue (BAT) is a highly metabolically active tissue [4]. Morphologically, BAT is densely packed with mitochondria and generates heat by short-circuiting the mitochondrial proton gradient via uncoupling protein 1 (UCP1), facilitating substrate use without ATP generation [5]. Thereby, active BAT significantly improves glucose and lipid clearance and raises energy expenditure [6,7]. Additionally, active BAT signals to other tissues improve the whole-body metabolic profile via the secretion of endocrine factors and miRNA containing exosomes [8,9]. In this regard, the recent demonstration of the presence of active BAT in adult humans has led to an increased interest in understanding the molecular signals underlying BAT differentiation and function [6,10].

The characterisation of the human transcriptome in the course of the ENCODE project revealed the pervasive transcription of three quarters of the human genome [11]. Most of the transcribed sequences however do not fall within protein-coding regions, but give rise to non-coding RNA, such as long non-coding RNA (lncRNA) [11]. lncRNAs are defined as non-coding genes giving rise to transcripts of more than 200 nt, which do not belong to an otherwise functionally defined class of RNA [12]. The lack of a functional definition coincides with a broad range of modes of function: lncRNAs have been shown to act both in cis as well as in trans [12,13], via an interaction of the transcribed RNA molecule with other RNA, proteins, or the DNA [14,15]. Compared to coding genes, lncRNAs are on average lower expressed, but show more tissue and developmental stage-specific expression profiles, advocating for a role as fine-tuning regulators of cellular function [16]. Selected lncRNAs have been shown to interfere with adipose tissue function and differentiation such as *lncBATE10*, which acts as a decoy for CELF1, which would otherwise bind to and repress *Pgc1a* mRNA [17], *H19*, which functions as a BAT-specific gatekeeper of paternally expressed genes [18], and *Ctcflos,* which regulates the expression and splicing of *Prdm16* [19]. However, their overall contribution to these processes remains elusive.

In this study, we performed RNA sequencing on BAT from C57BL/6 mice challenged with cold treatment and a high-fat diet, two physiologically relevant models of BAT activation [5,20], as well as on a set of seven metabolically active tissues. We found a set of adipose tissue-specific cold and/or diet-regulated lncRNAs, from which we selected *Gm15551* as a candidate for functional studies. The genomic locus of *Gm15551* is bound by PPARG and PRDM16 in brown adipocytes and it is upregulated in adipogenesis, and downregulated upon β-adrenergic stimulation in adipocytes. We performed the comprehensive phenotyping of gain and loss of function in vitro as well as a loss of function in vivo, but could not detect any phenotype related to *Gm15551*.

## 2. Material and Methods

### 2.1. Animal Experiments

Unless otherwise stated, mice were kept at 22 °C to 24 °C on a regular 12 h light cycle with ad libitum access to food and water. Wild type C57BL/6N mice used for the detection of differentially regulated genes and ChIP-Seq in interscapular brown adipose tissue (iBAT) were fed a chow diet (Ssniff V1554) up to the age of 8 weeks, where the respective cohorts were put on high-fat diet (HFD) (Ssniff D12492 (I) mod.) for 12 weeks and kept at 4 °C for 24 h. ΔGm15551 mice used for transcriptomics analyses were fed a chow diet (Altromin 1324). ΔGm15551 mice used for metabolic phenotyping were fed a chow diet (Altromin 1314) and respective cohorts were put on a high-fat diet (Ssniff D12492 (I) mod.) for 12 weeks starting from 8 weeks of age.

### 2.2. Generation of Gm15551 Knock out Animals

The mouse model for genetic deficiency of *Gm15551* was generated at the Czech Centre for Phenogenomics using CRISPR/Cas9 targeting exon 1 of *Gm15551* on the background of C57BL/6N. Sanger sequencing verified a 1113 bp deletion including exon 1 of *Gm15551* (3:126462197-126463309, GRCm38). Knock-out mice were backcrossed with wild-type animals for two generations to minimise the risk of off-target effects. Genotyping was carried out by two separate PCRs using primers F3/R2 (Appendix A; 372 bp and 1377 bp for knock out and wild type alleles respectively) and F5/R4 (234 bp for wildtype allele only). DNA extracted from tail tips using proteinase K digestion and Chelex 100 Resin (Bio-rad, Hercules, CA, USA) was amplified for 30 cycles at 95 °C, 60 °C, and 72 °C for 30 s each and visualised using capillary gel electrophoresis (Fragment Analyzer, Advanced Analytical Technologies, Ankeny, IA, USA).

### 2.3. Indirect Calorimetry

Prior to the experiment, a complete calibration protocol for the gas analysers was run according to the manufacturer’s recommendations, and the mice were weighed. The mice were singly housed in a PhenoMaster device (TSE systems, Bad Homburg, Germany) at a regular 12 h light cycle and 55% relative humidity with ad libitum access to water and the respective diet. At 11, 15, and 19 weeks of age, mice underwent a temperature challenge starting at 23 °C, followed by 6 h at 30 °C, 18 h at 4 °C, and 9 h at 23 °C again. The sampling rate was 15 min.

### 2.4. IPGTT

Animals were fasted overnight (16 h to 18 h) with free access to water. After weighing, the mice received 2 g/kg i.p. glucose. Blood glucose was measured fasted and after 15, 30, 60, and 120 min using a standard glucometer.

### 2.5. Adipocyte Diameter

Hematoxylin and eosin staining was performed on tissue slices and slides were scanned. Two representative areas per tissue were exported and analysed using adiposoft [21].

### 2.6. RNA Isolation and Reverse Transcription

Cells or frozen tissue samples were homogenised and lysed in TRIsure (Bioline, Modiin, Israel). Total RNA was isolated using EconoSpin All-In-One Mini Spin Columns (Epoch Life Science, Missouri City, TX, USA) and reverse-transcribed into cDNA using the High Capacity cDNA Reverse Transcription Kit (Applied Biosystems, Waltham, MA, USA), following the manufacturer’s instructions.

### 2.7. Quantitative Polymerase Chain Reaction (qPCR)

qPCR primer sets were designed to either include one primer overlapping an exon-exon junction, or, in case this was not possible, the two primers were placed in different exons to exclude the amplification of genomic DNA. qPCR was performed in 384 well format in a LightCycler 480 II (Roche, Basel, Switzerland). Moreover, 4 µL 1:20 diluted cDNA, 0.5 µL gene specific primer mix (5 µL each) and 4.5 µL FastStart Essential cDNA Green Master (Roche) were amplified using 45 cycles of 25 s at 95 °C, 20 s at 58 °C, and 20 s at 72 °C after 300 s at 95 °C initial denaturation. All combinations of primers and samples were run in duplicates and C_q_ values were calculated as the second derivative maximum. Genes of interest were normalised against housekeeper genes using the ΔC_q_ method. The primers used in this study can be found in Appendix A.

### 2.8. Total RNA Sequencing

RNA sequencing and library preparation were performed at the Cologne Center for Genomics (Cologne, Germany) according to their standard protocols. Before RNA sequencing, rRNA was depleted according to the instructions of the Illumina TruSeq kit. All sequencing experiments were accomplished with a paired-end protocol and a depth resulting in paired reads per sample. Before RNA sequencing, genomic DNA was eliminated following the instructions of the TURBO DNA-freeTM Kit, and subsequently, 1 µL RNA was used to examine RNA integrity in an Agilent 2100 Bioanalyzer (Agilent Technologies, Santa Clara, CA, USA).

### 2.9. Poly A RNA Sequencing

Paired end libraries were constructed using the NEBNext Ultra II RNA Library Prep Kit for Illumina (New England Biolabs, Ipswich, MA, USA) following the manufacturer’s protocol, and sequenced on a NovaSeq 6000 (Illumina, San Diego, CA, USA) in 2 × 50-bp paired-end reads.

### 2.10. RNA Sequencing Data Analysis

Reads were quality-filtered using cutadapt [22]. For visualisation, reads were mapped to the GRCm38 genome using STAR [23]. For quantification, reads were mapped to the Gencode M22 transcriptome or a combination of M22 and RNAcentral 5 using salmon [24].

### 2.11. ChIP Sequencing

For histone modification sequencing, brown adipose tissues of two mice were used in each sequencing experiment. Prior to ChIP, BAT was dissociated using a gentleMACSTM Dissociator (Miltenyi biotec, Bergisch Gladbach, Germany). The cell suspension was cross-linked with 1% formaldehyde for 10 min at RT and the reaction was quenched with 0.125 M glycine at RT. Cells were washed twice with cold PBS and PMSF and snap-frozen in liquid nitrogen before storing at −80 °C.

Frozen pellets were thawed on ice for 30 min to 60 min. Pellets were resuspended in 5 mL lysis buffer 1 (50 nM Hepes, 140 mM, 1 EDTA, 10% glycerol, 0.5% NP-40, 0.25% Triton X-100) by pipetting and then rotated vertically at 4 °C for 10 min. Pellets were resuspended in 5 mL lysis buffer 2 (10 mM Tris, 200 mM, 1 mM EDTA, 0.5 mM EGTA) and incubated at vertical rotation and at room temperature for 10 min. Samples were centrifuged for 5 min at 1350× *g* at 4 °C and supernatant was carefully aspirated. Then, samples were resuspended in 3 mL lysis buffer 3 (10 mM Tris, 100 mM NaCl, 1 mM EDTA, 0.5 mM EGTA, 0.1%-deoxycholate, 0.5% N-lauroylsarcosine) and were separated into 2 times 1.5 mL in 15 mL polypropylene tubes, in which they were sonicated with the following settings by Bioruptor^®^ Plus (Diagenode, Seraing, Belgium) sonication: power = high, on interval = 30 s, off interval = 45 s, total time = 10 min (18 cycles of on/off). Sonicated samples were transferred to a 1.5 mL microfuge tube and were centrifuged for 10 min at 16,000× *g* at 4 °C to pellet cellular debris. Furthermore, 10% of sample solution was stored to be used as input control, while the rest was used for ChIP.

To capture different histone modifications, 5 µg to 10 µg of the respective antibodies (Appendix A) were added to the sonicated ChIP reaction and rotated vertically at 4 °C overnight. Moreover, 100 µL Dynabeads (Protein A or Protein G) for each ChIP sample were prepared according to the manufacturer’s instructions, mixed with 1 mL of antibody-bound chromatin and rotated vertically at 4 °C for at least 2 h to 4 h. Bound beads were washed at least five times in 1 mL cold RIPA (50 mM Hepes, 500 mM LiCl, 1 mM EDTA, 1% NP-40, 0.7% Na-deoxycholate) and once in 1 mL cold TE buffer containing 50 mM NaCl. Samples were eluted for 15 min with elution buffer (50 mM Tris, 10 mM EDTA, 1% SDS) at 65 °C and continuously shaken at 700 min^−1^. Beads were separated using a magnet and 200 µL supernatant were transferred to fresh microfuge tubes. Input samples were thawed and mixed with 300 µL elution buffer. ChIP/input samples were incubated at 65 °C in a water bath overnight to reverse the cross linking reaction. TE buffer was added at room temperature to dilute SDS in both ChIP and input samples. For digestion of RNA and protein contamination, RNase A was added to the samples and incubated in a 37 °C water bath for 2 h; then proteinase K was added to a final concentration of 0.2 mg/mL and incubated in a 55 °C water bath for 2 h. Finally, DNA was extracted using a standard phenol-chloroform extraction method at room temperature and DNA concentrations were measured using a NanoDrop ND-1000 spectrophotometer (Thermo Fisher Scientific, Waltham, MA, USA) or Qubit dsDNA HS Assay Kit (Thermo Fisher Scientific) and stored at −80 °C until sequencing.

### 2.12. ChIP Sequencing Data Analysis

Reads were mapped to the GRCm38 genome using bowtie2 [25] after quality filtering by cutadapt. Peaks were called using macs2 compared to the corresponding input sample (−c flag).

### 2.13. Tissue Specificity

Tissue specificity scores were calculated for every gene over seven metabolically active tissues as log_2_(TPM_n_ + 1)/Σlog_2_(TPM + 1), as described in [26] from RNA-Seq data that we have previously published (GEO: GSE121345 [27]).

### 2.14. Gene Set Enrichment Analysis

Gene set enrichment was done using topGO [28] for GO and ReactomePA [29] for reactome.

### 2.15. Assessment of Coding Potential

Coding potential was calculated using CPAT [30]. Ribosome scores were calculated as log2(TRAP/totalRNA), where TRAP is the normalised counts from a publicly available dataset of TRAP of mouse iBAT (GEO: GSE103617) and totalRNA are the normalised counts from the room temperature control diet iBAT total RNA samples.

### 2.16. Primary Cell Culture

Inguinal and epididymal white as well as intrascapular brown adipose tissues from 6- to 8-week-old C57BL/6J mice were dissected, minced, and digested with collagenase II (Worthington Biochemical Corporation, Lakewood, NJ, USA) and dispase II (Sigma-Aldrich, St. Louis, MO, USA; iBAT only). Cells were seeded in 24 well plates and grown in DMEM/Ham’s F12 medium, supplemented with 0.1% Biotin/D-Pantothenate (33/17), 1% penicillin-streptomycin, and 20% FBS. Upon reaching confluency, FBS concentration was reduced to 10% and differentiation was induced using 1 µM rosiglitazone, 850 nM insulin, 1 µM dexamethasone, 250 µM 3-isobutyl-1-methyl-xanthin (IBMX), 125 µM indomethacine (brown only), and 1 nM T3 (iBAT only). Subsequently, the medium was changed every other day for medium containing 10% FBS, rosiglitazone, and T3 (iBAT only). Full differentiation was reached 7 days after induction. Cells were stimulated using 1 µM isoproterenol or 10 µM CL316243.

### 2.17. Cultivation of Brown Adipocyte Cell Lines

The cells were grown in high-glucose DMEM supplemented with 10% FBS and 1% penicillin-streptomycin. After reaching confluency, differentiation was induced by 0.5 µM rosiglitazone, 1 nM T3, 1 µM Dexamethasone, 850 nM insulin, 125 µM indomethacine, and 500 µM IBMX. Two days later, medium was exchanged for medium supplemented with 0.5 µM rosiglitazone and 850 nM insulin. Afterwards, medium was changed for medium containing 0.5 µM rosiglitazone every second day. Full differentiation was reached 7 days after induction.

PIBA cells were cultured in the same medium as wt1-SAM cells. A common induction/stimulation cocktail consisting of 10 µM rosiglitazone, 1 nM T3, 0.5 µM Dexamethasone, 850 nM insulin, 12.5 µM indomethacine, and 125 µM IBMX was used to differentiate cells.

### 2.18. In Vitro Gain and Loss of Function Studies

For in vitro gain of function studies using the wt1-SAM cell line, sgRNAs were designed using CRISPick [31] and cloned into the sgRNA (MS2) cloning backbone (addgene 61424), as described in [32]. The target sequences used are found in Appendix A.

LNA gapmers designed and synthesized by Qiagen were used for the in vitro loss of function experiments. Two non-targeting scrambled LNAs were used as control (Appendix A).

Preadipocytes were transfected by seeding 30,000 cells per well of a 24-well plate in 500 µL growth medium and adding 1.5 µL TransIT-X2 (Mirus) and 125 ng plasmid DNA or 1.4 µL LNA (10 µM) in 50 µL Opti-MEM I once the cells had attached.

In order to transfect mature adipocytes, 3 µL TransIT and 250 ng plasmid DNA or 1.4 µL (10 µM) in 100 µL Opti-MEM I were pipetted into the well of a 24 well plate. After 15 min, cells resuspended in 500 µL Opti-MEM were added. Moreover, 24 h later, the medium was changed to a regular differentiation medium.

### 2.19. Oil Red O Staining

Cells were fixed with 4% formalin for 30 min, and rinsed once with water followed by 60% isopropanol. Cells were stained with Oil Red O (0.3% in 60% isopropanol) for 10 min. Excess dye was rinsed with water. For quantification, the Oil Red O was eluted in 100% isopropanol and OD measured at 520 nm in a multi plate reader.

### 2.20. Statistical Analysis

Statistics for RNA-Seq data was done in DESeq2 [33] using LRTs for factors with multiple levels or for analysing multiple factors at once and wald tests otherwise. Log fold changes were shrunken and *s*-values calculated using apeglm [34]. Data from animal experiments with repeated measurements were averaged over temperature and day/night conditions and analysed using mixed-effects models with the body weight as a cofactor and individual animal as a random variable. Other data were analysed using Student’s *t*-tests and adjusted for multiple testing using Holm’s method. Shown are individual values in addition to the mean standard error of the mean. * means *p* < 0.05, ** means *p* < 0.01, *** mean *p* < 0.001.

## 3. Results

### 3.1. Total RNA-Seq Identifies lncRNAs Regulated in Activated iBAT

In order to identify lncRNAs implicated in the regulation of iBAT function, we set out to perform total RNA-seq on C57BL/6N mice put on a high-fat diet (HFD) regime from 8 weeks of age onwards for 12 weeks and additionally housed at 4 °C for 24 h at the end of this period (Figure 1A). We found that cold treatment significantly reduced the weight of the iBAT in both the control and the HFD group (Appendix A), while HFD treatment alone did not induce significant changes in iBAT weight. On the other hand, epididymal white adipose tissue (eWAT) and inguinal white adipose tissue (iWAT) as well as liver weights were increased upon HFD treatment, independent of the cold treatment. Cold treatment alone only induced an increase of eWAT but not iWAT and liver weight. Gene expression measurements reflected the observed difference in the reaction of white and brown adipose tissue to the treatments (Appendix A). Cold treatment induced a robust induction of the common adipose marker gene *Elovl3* as well as the brown adipose markers *Cidea*, *Dio2* and *Ucp1* while HFD alone was insufficient for the induction of any significant changes in those genes, although *Ucp1* was tendentiously upregulated. In iWAT however, cold and HFD treatment showed opposing effects. The white adipose marker gene Leptin (*Lep)* was tendentiously repressed upon cold treatment and induced by HFD while *Elovl3*, *Dio2* and *Ucp1* were upregulated by the cold and downregulated by the HFD treatment. The treatment regimes also directly affected the animals’ metabolism as seen by the significantly impaired glucose tolerance upon HFD treatment (Appendix A). Energy expenditure was elevated by cold treatment but reduced by HFD treatment (Appendix A) while the respiratory exchange rates indicated a shift towards lipid catabolism induced by both cold and HFD treatment (data not shown). Together, these data indicate our dataset is an adequate model for different functional states of iBAT. Additionally, we used a second dataset consisting of seven metabolically active tissues (iBAT, iWAT, eWAT, liver, kidney, muscle, heart) which we have generated for a previous study to be able to assess transcriptome-wide tissue specificity (GEO: GSE121345 [27]). To generate a comprehensive set of lncRNA genes, we combined the annotated transcript isoforms from GENCODE and the lncRNA isoforms from RNAcentral at the gene level.

Total RNA-seq of the iBAT dataset identified differentially expressed genes, of which 216 were lncRNA genes (likelihood ratio test (LRT), *p* < 0.001; Figure 1B; Appendix A). The largest cluster (cluster 3) consisted of genes that were induced by cold treatment independent of the diet and was enriched for genes involved in stress response and mitochondrion organisation (Figure 1C and Appendix A). Similarly, cluster 4 contained genes downregulated by cold treatment independently of diet and was enriched for gene expression regulation and signal transduction. The other clusters included genes with the synergistic interaction of HFD and housing temperature; either being induced (cluster 5, enriched for signalling) or repressed by HFD and cold treatment (clusters 1 and 2, enriched for extracellular matrix as well as metabolism). We used the transcriptomics data from the seven metabolically active tissues to calculate an adipose tissue enrichment score, defined as the ratio of log adipose tissue counts over the log of total counts (Appendix A). As temperature influenced the gene expression of more genes than HFD, we focused on genes regulated by the cold treatment. Wald tests identified 110 cold regulated lncRNA genes (*s* < 0.05; Appendix A), of which 65 (59%) also showed an adipose tissue-specific expression profile (adipose score > 50%; Figure 1D and Appendix A). On the other hand, from the 610 cold regulated coding genes, only 35% showed adipose tissue-specific expression (Appendix A). Notably, our analysis identified known brown adipose marker genes such as *Ucp1* and *Adcy3*, as well as the lncRNA genes *LncBate10* and *Ctcflos*, which have previously been shown to play a role in the regulation of brown/beige adipose tissue function [17,19], proving the applicability of our strategy towards the identification of novel candidate adipose regulating lncRNAs. In order to rule out that any of the identified lncRNA genes were differentially regulated because of an increased immune cell infiltration of the iBAT caused by the cold or HFD treatment [20], we checked the expression profiles of several immune cell marker genes [35], of which none were differentially regulated (Appendix A).

### 3.2. Gm15551 Is an Adipose Specific, Highly Regulated lncRNA

We further evaluated the identified candidate lncRNA genes towards their potential relevance for the regulation of adipose tissue function by checking for binding sites of transcription factors relevant for adipose tissue. As we were specifically interested in investigating independently transcribed lncRNA genes, we excluded lncRNAs bidirectionally transcribed from a shared promoter with a coding gene, as well as antisense lncRNAs overlapping exonic sequence from a coding gene. We finally focused our study on the lncRNA *Gm15551*, which was highly adipose-specific and significantly repressed upon cold treatment in iBAT (Figure 1D). While HFD alone was not sufficient to induce the repression of *Gm15551* expression, the combination of cold treatment and HFD further repressed *Gm15551* compared to cold treatment alone (ANOVA; *p* = 0.00070, Figure 2A). Among the examined tissues, the expression of *Gm15551* was observed to be strictly restricted to adipose tissue, such as the common adipocyte marker genes *Adipoq* and *Pparg* (Figure 2B). Within the three adipose tissues that we looked at, *Gm15551* showed the highest expression in eWAT and the lowest in iBAT, with intermediate expression in iWAT, in anti-correlation with the expression of the thermogenic adipocyte marker gene *Cidea*. The notion of this expression pattern together with the repression upon activation of thermogenesis in iBAT led us to hypothesize that *Gm1555* might have an anti-thermogenic function.

*Gm15551* is expressed from chromosome 3 and there are 2 transcripts annotated which both consist of 2 exons and only differ in the exact position of the transcription end site (Figure 2C). An analysis of publicly available ChIP-Seq data showed that the locus is bound by the core thermogenic transcription factor PRDM16 in iBAT. Additionally, we found that PPARG binds to the *Gm15551* locus in eWAT, iWAT, as well as iBAT, with the height of the ChIP-Seq peak correlating with the *Gm15551* RNA expression levels. Chromatin features such as the ratio of H3K4me1 relative to H3K4me3 have previously been used to distinguish promoters from enhancers [36]. Therefore, we looked at histone modifications using ChIP-Seq (Appendix A). We found higher levels of H3K4me3 compared to H3K4me1, which is indicative of a promoter as opposed to an enhancer [36]. The H3K27ac signal was higher than H3K4me3 or H3K4me1, and H3K27me3 was basically absent.

As *Gm15551* is expressed antisense from a locus within intron 2 of the intracellular Ca^2+^ signalling protein *Camk2d* and it is known that lncRNAs can work as in cis regulators of nearby coding genes [12], we checked whether the expression of *Gm15551* correlates with the expression of *Camk2d* in various publicly available RNA-Seq datasets of adipose tissue, but found no significant correlation (ANCOVA, *p* = 0.848; Appendix A). To exclude the possibility of *Gm15551* being a coding gene wrongly annotated as lncRNA [37], we calculated the coding potential for all genes expressed in our dataset using CPAT [30]. *Gm15551* showed a low coding probability comparable to other known lncRNA genes, as opposed to the coding brown adipocyte marker genes *Cidea*, *Adcy3*, and *Ucp1* (Figure 2D). Similarly, ranking genes by the ratio of ribosome associated over total RNA in a publicly available TRAP-Seq data set of iBAT sorted *Gm15551* with other lncRNA genes (Appendix A).

Next, we followed the gene expression of *Gm15551* during the differentiation of preadipocytes into mature adipocytes. Our analysis showed that *Gm15551* is highly upregulated early in differentiation, like the core adipocyte transcription factor *Pparg*, and unlike *Ucp1*, which only reaches maximum levels in late differentiation (Figure 2E). To mimic the effects of cold treatment on adipose tissue in vitro, we stimulated cells with the non-selective β-adrenergic agonist isoproterenol or β_3_-specific agonist CL316243. Both stimuli were sufficient to repress *Gm15551* in differentiated adipocytes originating from eWAT, iWAT, and iBAT (Figure 2F). In primary immortalized brown adipocytes, the effect of β-adrenergic stimulation on the expression of *Gm15551* was stable over 24 h (Appendix A).

### 3.3. Gain- and Loss-of-Function of Gm15551 Does Not Disturb Brown Adipocyte Development and Function In Vitro

To investigate the role of *Gm15551* in the differentiation and function of brown adipocytes, we used an immortalized brown preadipocyte cell line stably expressing the CRISPRa SAM system for gain-of-function studies, together with locked nucleic acid (LNA) antisense oligonucleotides for loss-of-function studies [38]. The transfection of either one of two plasmids encoding single guide RNAs (sgRNAs) targeting *Gm15551* two days prior to the induction of the differentiation led to a robust overexpression of *Gm15551* compared to the empty vector control at day 1 of differentiation (Figure 3A). The effect of the overexpression was greatly diminished on days 4 and 7 because the natural gene expression of *Gm15551* rises during differentiation. However, we could not observe any changes in the expression of common and brown adipocyte marker genes or in the cells’ ability to accumulate lipids (Figure 3A,B and Appendix A).

Next, we set out to knock down *Gm15551* in mature adipocytes. To detect potential interactions of *Gm15551* expression with the thermogenic activation of brown adipocytes, we looked at cells both under basal conditions and under β-adrenergic stimulation. Reverse transfection with two different LNAs targeting *Gm15551* on day 4 of differentiation resulted in the robust downregulation of *Gm15551* on day 7 compared to the non-targeting control LNA (Appendix A). Overall, we found 2762 genes differentially regulated by either knockdown or stimulation (Figure 3C). Hierarchical clustering showed that the influence of the β-adrenergic stimulation was more pronounced than that of the loss-of-function of *Gm155551*. Samples treated with the control non-targeting LNA clustered together with LNA1 treated samples in both the basal and stimulated condition, indicating that the two LNAs used caused different effects. Looking at the specific effect of each LNA individually, we found 70 and 188 differentially regulated genes, respectively (Figure 3E and Appendix A). There was only an overlap of 16 genes detected to be significantly regulated by the knockdown of *Gm15551* using either of the two LNAs. A GO analysis revealed that these genes were enriched for signalling and especially NF-κB mediated signalling (data not shown).

Similarly, reverse transfection with sgRNA encoding plasmids led to a small but significant overexpression of *Gm15551* in fully mature adipocytes and notably, was able to suppress its downregulation upon β-adrenergic stimulation (Appendix A). However, we did not observe any changes in the gene expression of any of the probed adipocyte marker genes. We further raised the overexpression efficiency by the simultaneous transfection of two different plasmids encoding sgRNAs targeting *Gm15551* (Appendix A). We sequenced the transcriptomes of these samples, and found a total of 792 genes differentially regulated by either gain-of-function of *Gm15551* or β-adrenergic stimulation (Figure 3D). The effect of the thermogenic activation dominated the dataset, as shown by hierarchical clustering. However, there were also two clusters with genes affected by the overexpression of *Gm15551*. When we specifically looked for changes in gene expression caused by the *Gm155551* gain-of-function, we found 14 differentially expressed genes and GO analysis showed an enrichment for genes involved in inflammatory response (Figure 3E,F).

A comparison between the genes differentially regulated by the gain and loss-of-function of *Gm15551* showed that there was no overlap. Additionally, most genes affected by the knock down of *Gm15551* showed no change in gene expression in the gain-of-function experiment (Appendix A). Finally, the Oil red O staining of mature adipocytes showed no effect of either gain or loss-of-function of *Gm15551* on the cells’ ability for lipid accumulation (Appendix A).

### 3.4. Gm15551 Loss-of-Function Does Not Impair Adipose Tissue Function In Vivo

Next, we created a loss-of-function mouse model by knocking out the exon 1 of *Gm15551*. Since brown adipose tissue plays a role in the regulation of body weight as well as lipid and glucose metabolism [39], we challenged homozygous ΔGm15551 mice and wild type litter mates from 8 weeks of age for 12 weeks with a high fat diet and repeatedly measured body weight and performed intraperitoneal glucose tolerance test (IPGTT), and indirect calorimetry (Appendix A). While HFD was sufficient to provoke a significant raise in body weight (Figure 4A; *p* = 0.0077) characterised by an increased amount of body fat (Appendix A; *p* = 0.00014), we did not observe significant changes induced by the loss-of-function of *Gm15551* (*p* = 0.067 and 0.56 respectively). Similarly, prolonged HFD treatment but not *Gm15551* loss-of-function resulted in impaired glucose tolerance (Figure 4B; *p* = 0.0043 and 0.80 respectively). Additionally, the adipocyte diameter and morphology of HFD-treated animals were not affected by *Gm15551* knockout (Appendix A; *p* = 0.36). Next, we performed indirect calorimetry while sequentially changing the temperature first from room temperature to thermoneutrality (30 °C), followed by a period at 4 °C before returning to room temperature (23 °C). Upon the beginning of thermoneutrality, energy expenditure slightly dropped and consequently raised when the temperature was dropped. Upon return to room temperature, the energy expenditure went back to the starting point (Figure 4C). However, there was no effect of the *Gm15551* knockout (*p* = 0.66). The respiratory exchange ratio of control diet animals raised with the onset of the first dark phase at thermoneutrality and dropped again in the following light phase, indicating the combustion of carbohydrates taken up with the food during the dark phase (Figure 4D). With prolonged cold treatment, the respiratory exchange rate rose again to an intermediate value, indicating that mice had to take up food in addition to combusting stored lipids. At the end of the cold treatment, the respiratory exchange rate rose even further, indicative of the mice mostly relying on energy from the ingested carbohydrates. This effect of temperature and day light cycle was mostly suppressed in HFD animals, as they take up less carbohydrates with their alimentation. Again, there was no evidence of an impact of the *Gm15551* loss-of-function (*p* = 0.95). 

When we compared the adipose tissue transcriptomes from HFD and control diet-fed animals, we found 5655 differentially expressed genes (LRT, *p* < 0.001; Figure 4E). Hierarchical clustering was strongly driven by the difference between brown and white adipose tissues. Gene ontology analysis showed that the genes with higher expression in brown adipose tissue were enriched for the terms related to mitochondria, while genes showing a higher expression in white adipose tissue were enriched for terms related to immune system and locomotion (Appendix A). The direct comparison of the samples from wild type animals with those from knock out animals revealed only 11 differentially regulated genes (wald test, *s* < 0.05; S4F). Additionally, we compared the gene expression for iBAT from wild type and ΔGm15551 mice, both at room temperature and after 24 h of cold treatment. Overall, there were 2531 differentially expressed genes (LRT, *p* < 0.001; Appendix A), which clustered the samples by temperature, but not genotype (Figure 4F). Both sets of up- and down-regulated genes upon cold treatment showed enrichment for terms related to metabolism (Appendix A). Furthermore, the direct comparison of the wild type transcriptomes with those from ΔGm15551 animals only revealed six differentially expressed genes (Wald test, *s* < 0.05; Appendix A; Appendix A), while the cold treatment affected 1864 genes (Wald test, *s* < 0.05; Appendix A).

## 4. Discussion

Cold and nutrient-activated BAT regulates energy homeostasis and improves metabolic status via non-shivering thermogenesis and the secretion of endocrine factors [6,8]. LncRNAs have been shown to be tissue-specific fine-tuning regulators of tissue function, and therefore have been proposed as potential selective targets for the treatment of different diseases [40,41]. In recent years, the function of some lncRNAs expressed in adipose tissue has been described (reviewed by [42]). However, the function of most lncRNAs remains unknown. Here, we detected a set of 65 lncRNAs of which the expression is specific to adipose tissue and correlates with BAT function and characterised *Gm15551* further in vitro and in vivo.

We found *Gm15551* to be highly adipose tissue-specific, with a higher expression in white compared to brown adipose tissues. *Gm15551* is highly induced in the early stages of brown adipogenesis and downregulated upon β-adrenergic stimulation in both white and brown adipocytes. We could show that the key transcription factor PRDM16, which controls the determination of brown adipocytes and the browning of white adipose tissue [43,44], binds to the *Gm15551* locus in iBAT. Furthermore, we found the *Gm15551* locus to be occupied by PPARG in white as well as brown adipose tissues with a more pronounced occupancy in white compared to brown adipose tissue. PPARG is a transcription factor involved in the maintenance of the general adipocyte phenotype, but also showing depot-specific binding patterns [45]. These findings led us to hypothesize an adipocyte-specific function of *Gm15551*.

Previous studies have shown that lncRNAs might give rise to unidentified translation products [37,46]. We used a sequence-based bioinformatics tool to calculate coding probability and analysed a public TRAP-Seq data set to detect ribosome associated RNAs. Our results showed that *Gm15551* has a low coding probability and is not associated with ribosomes. While enhancers are known to give rise to bidirectionally transcribed, short, unspliced, and unstable enhancer RNAs (eRNAs), it has recently been reported that some enhancers can also be the place of unidirectional transcription giving rise to spliced lncRNAs [36,47]. The *Gm15551* locus featured a low ratio of the H3K4me1 over the H3K4me3 histone mark, indicative of promoters. However, the *Gm15551* locus also features H3K27ac histone marks, high levels of which are characteristic for enhancers [36]. Enhancers are cis-regulatory elements, regulating the expression of nearby genes. Therefore, we compared the expression of *Gm15551* and *Camk2d*, which overlaps with *Gm15551* in the genome, in several adipocyte-related RNA-Seq datasets, but found no correlation. However, we cannot rule out a potential enhancer function of *Gm15551*.

In order to unveil the potential effects of *Gm15551* gain-of-function on brown adipogenesis, we overexpressed *Gm15551* two days prior to the induction of differentiation in a brown preadipocyte cell line, but could not measure any effect of the overexpression neither on common and brown adipocyte marker genes nor on lipid accumulation, both under basal conditions and under β-adrenergic stimulation. Likewise, there was no effect on lipid accumulation in the subsequent gain- and loss-of-function experiments in mature brown adipocytes. At the transcriptome level, we hypothesised that gain- and loss-of-function of *Gm15551* should lead to opposite effects on the gene expression of potential target genes of *Gm15551*. However, there were no genes that were significantly regulated in both datasets. Furthermore, most genes differentially regulated in one dataset did not even show a (non-significant) regulation in the other one. Those genes that were oppositely regulated by *Gm15551* gain- and loss-of-function such as *Lcn2*, *Saa3*, and *Hp* are inflammatory markers [48,49,50]. We have previously found them to be differentially regulated by other sgRNAs in several datasets using the wt1-SAM model system and therefore interpret them as a model-specific artefact.

As impaired iBAT function has been shown to render mice susceptible to diet-induced obesity and insulin intolerance [51,52], we put ΔGm15551 mice on HFD for 12 weeks and additionally repeatedly tested their response to cold treatment by indirect calorimetry. While both the prolonged HFD treatment and sex caused clear differences, we could not detect any significant effect of the *Gm15551* loss-of-function on the examined adipose tissue and metabolic parameters, such as energy expenditure, body weight, and glucose tolerance. When we analysed adipose tissue transcriptomes, clear differences between the white and brown adipose tissues as well as between iBAT from cold treated and control animals became evident. However, the *Gm15551* knockout only caused a minor number of differentially expressed genes, not exceeding what is expected as false positives.

We have identified a set of adipose tissue-specific, HFD, and cold regulated lncRNAs from which we characterised *Gm15551*. Although it is highly upregulated during brown adipogenesis and its expression correlates with iBAT activity, we could not detect a phenotype of either gain- or loss-of-function of *Gm15551* in vitro. Likewise, we could not detect a *Gm15551*-related phenotype when performing comprehensive transcriptomic and adipose tissue physiologic phenotyping in vivo. In summary, our findings indicate that Gm15551 is dispensable for iBAT development and function, despite its marked upregulation during initial adipose tissue development. This result is in concordance with a study which previously found Gm15551 to be upregulated in both white and brown adipogenesis but detected no effect of the siRNA-mediated knockdown of Gm15551 on white adipocyte differentiation [53]. While we have not ruled out a potential effect of a knockdown of Gm15551 in brown preadipocytes on adipogenesis, the lack of a phenotype in white adipogenesis as well as in vivo investigations makes it appear implausible to find a phenotype in brown adipogenesis. A functional redundancy has been reported for coding genes such as CD34 and for duplicated genes in general [54,55]. Furthermore, lncRNAs have been shown to have tissue and cell state dependent and potentially very subtle functions [16]. Our combined in vitro and in vitro approach has not shown any effect of either gain or loss of function of Gm15551 on adipocyte differentiation and gene expression, and has shown that Gm15551 loss of function does not affect adipose tissue specific physiological parameters such as body weight and energy expenditure even when the mice are exposed to cold or HFD, two stressors of adipose tissue. However, we cannot rule out the possibility of the existence of other genes showing functional redundancy to Gm15551, hiding any effects of the Gm15551 loss-of-function. Furthermore, we cannot rule out that the potential regulatory role of Gm15551 is below the threshold to be detected in our measurements or only becomes relevant under very specific circumstances which have not been investigated in this study. Recently, two publications investigated the role of sets of lncRNAs selected based on high expression levels, conservation or genomic proximity to known coding developmental regulatory genes in mouse and zebra fish respectively [56,57]. Interestingly, both studies fail to detect a phenotype for the majority (31 out of 32 and 11 out of 12) of the investigated lncRNAs. The authors attribute this partially to the robustness of the embryonal developmental process and the proposed role of lncRNAs as fine tuners of biological processes as compared to being master regulators [57]. The advent of short-read sequencing allowed for the detection of pervasive transcription of large parts of mammalian genomes. This led to the annotation of large numbers of novel lncRNAs, far outnumbering functional studies investigating these genes. Even though evidence accumulates for the functionality of a rising number of lncRNAs, the ratio between functional and ‘junk’ lncRNA genes remains a matter of debate [58].

In conclusion, while *Gm15551* is specifically expressed in adipose tissues and its expression is significantly regulated following cold stimulation, and although we subjected mice both to HFD and cold treatment, two major stressors of adipose tissue [20,59], it is possible that *Gm15551* (I) exhibits either a very subtle function undetectable by our measurements, (II) a context-dependent function in a specific cellular state that we have not investigated, or (III) that *Gm15551* has no biological function in murine adipose tissue.

## Figures and Tables

**Figure 1 ncrna-08-00032-f001:**
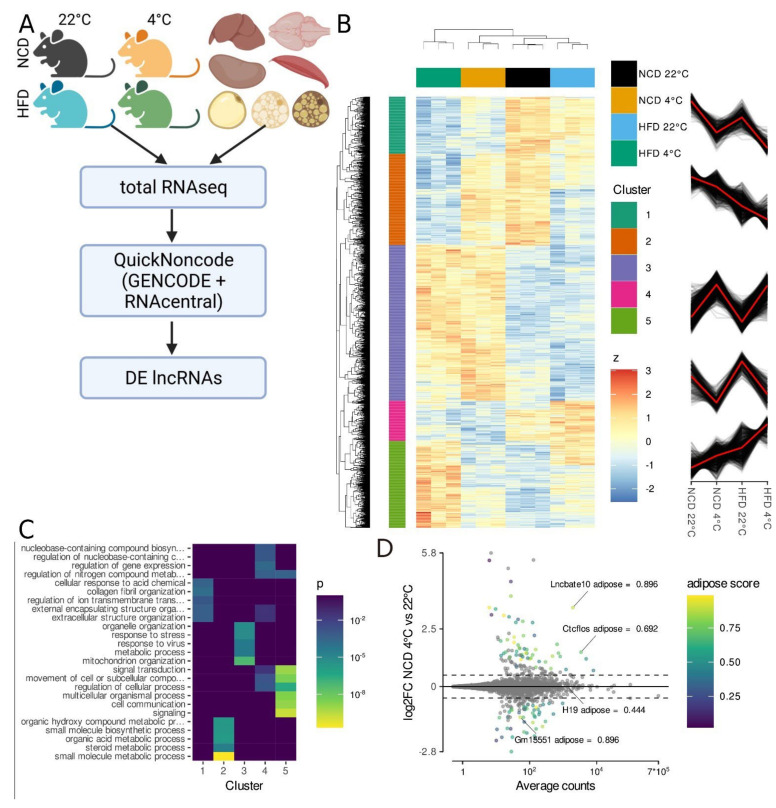
RNA-Seq reveals temperature and obesity dependent changes in iBAT lncRNA expression. (**A**) Experimental design. Total transcriptomes from the interscapular brown adipose tissue (iBAT) of 20-week-old mice housed at 22 °C or 4 °C for 24 h (*n* = 3) fed either a high-fat (HFD) or a control diet (NCD) for 12 weeks were analysed together with the total-transcriptomes from seven metabolically active tissues (*n* = 1, GSE121345). The union of GENCODE and RNAcentral annotated genes was used for the analysis to reveal which are both adipose tissue specific and regulated by physiologically relevant stimuli. Created with BioRender.com (the last access on 10 April 2022). (**B**) Hierarchical clustering of genes differentially regulated by diet and cold treatment in adipose tissue (likelihood ratio test (LRT), FDR < 0.001). Colour code depicts row-wise standardised expression. (**C**) Gene ontology (GO) enrichment analysis for the gene clusters shown in (**B**). (**D**) Expression levels and changes for long noncoding RNA (lncRNA) genes in iBAT from cold treated compared to control mice on control diet. Genes showing significant differential gene expression are colour-coded, indicating their adipose tissue specificity (Wald test, log2FC > 0.5, *n* = 6, s < 0.05).

**Figure 2 ncrna-08-00032-f002:**
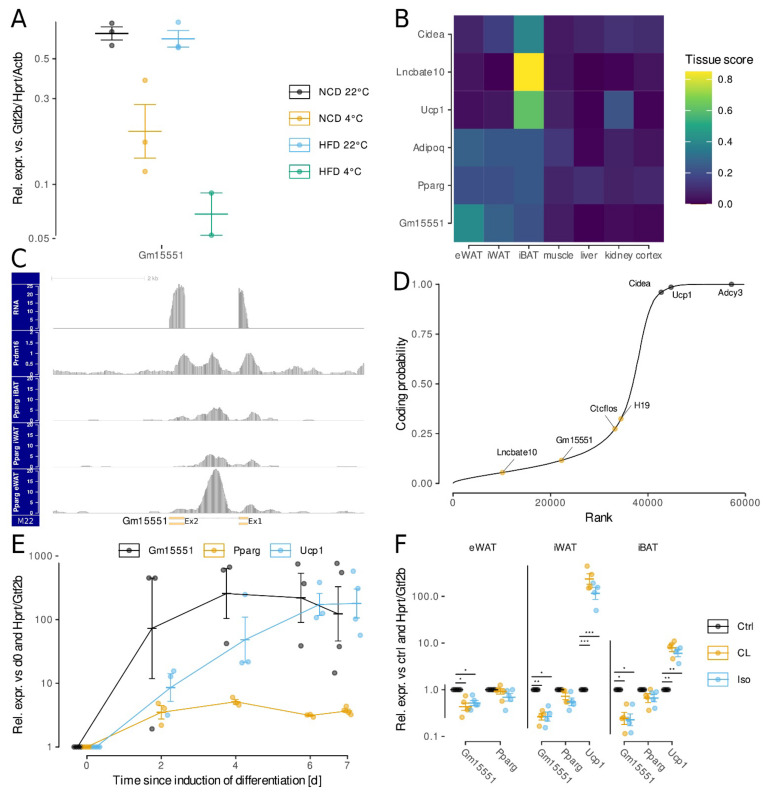
Gm15551 is an adipose tissue-specific, diet- and temperature-regulated lncRNA. (**A**) Expression of *Gm15551* in iBAT from cold treated and/or HFD fed mice. (**B**) Expression profile of Gm15551, the common adipocyte marker genes *Pparg* and *Adipoq* as well as the iBAT specific lncRNA *Lncbate10* and the brown adipocyte marker genes *Ucp1* and *Cidea* in seven metabolically active tissues. (**C**) Genomic locus of *Gm15551* showing the RNA expression as well as binding sites of PRDM16 in iBAT (GEO: GSE63965) and PPARG in eWAT, iWAT and iBAT (GEO: GSE41481) as well as the annotated transcripts in GENCODE M22. (**D**) Ranked coding probability of all genes expressed in the dataset as calculated by CPAT. Indicated are *Gm15551*, the coding genes *Ucp1*, *Adcy3* and *Cidea* as well as the lncRNAs *Ctcflos*, *H19* and *LncBate10*. (**E**,**F**) Expression profiles of *Gm15551*, the common adipocyte marker gene *Pparg* and the brown adipocyte marker gene *Ucp1* during the differentiation of PIBA cells (**E**) and in fully differentiated primary adipocytes (**F**) stimulated for 24 h with the non-selective β-adrenergic agonist isoproterenol or the β_3_-specific agonist CL316243 (paired *t*-test, *n* = 4–5). * means *p* < 0.05, ** means *p* < 0.01, *** mean *p* < 0.001.

**Figure 3 ncrna-08-00032-f003:**
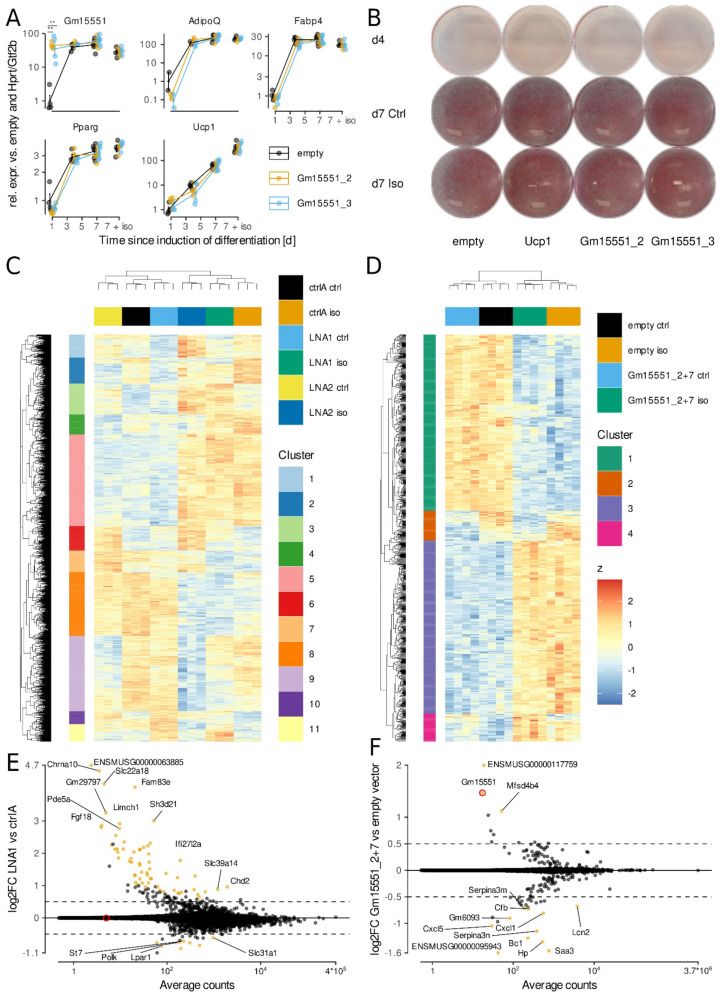
Gm15551 is dispensable for iBAT function in vitro. (**A**,**B**) Gene expression profiles (**A**) of *Gm15551*, the common adipocyte marker genes *Pparg*, *Adipoq* and *Fabp4* as well as the brown adipocyte marker gene *Ucp1* (paired *t*-test, *n* = 2–6) and oil red O staining (**B**) of wt1-SAM cells transfected with plasmids coding for sgRNAs targeting *Gm15551*, *Ucp1* or empty vector two days before induction of differentiation. (**C**,**D**) Hierarchical clustering of genes differentially regulated by β-adrenergic stimulation and/or knockdown (**C**) or overexpression (**D**) in mature adipocytes at day 4 of differentiation (LRT, *n* = 3 (**C**) or 4 (**D**), *p* < 0.001). (**E**,**F**) Effect of knockdown (**E**) or overexpression (**F**) of Gm15551 on gene expression in mature wt1-SAM cells. The log2FC is the average over the effect in isoproterenol stimulated and control cells (wald test, log2FC > 0.5, *n* = 6 (**E**) or 8 (**F**), s < 0.05).

**Figure 4 ncrna-08-00032-f004:**
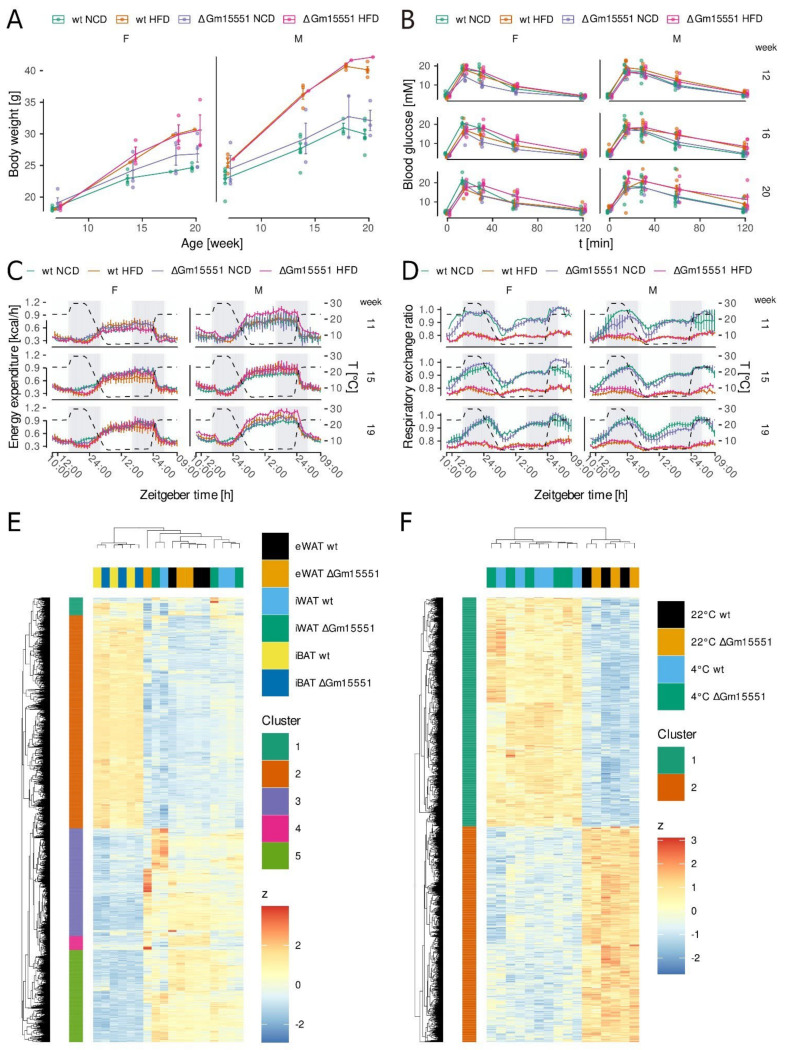
Gm15551 is dispensable for iBAT function in vivo. (**A**) Body weight of ΔGm15551 and wild type mice fed a high fat or control diet. (**B**) Intraperitoneal glucose tolerance test (IPGTT) of 12-, 16-, and 20-week-old ΔGm15551 and wild type mice fed HFD or NCD. (**C**,**D**) Energy expenditure (**C**) and respiratory exchange rates (**D**) of 12-, 16-, and 20-week-old ΔGm15551 and wild-type mice fed HFD or NCD. (**E**) Hierarchical clustering of genes differentially regulated between adipose tissues or by knockout of Gm15551 in 12-week-old mice (LRT, *n* = 3, *p* < 0.001). (**F**) Hierarchical clustering of genes differentially regulated by temperature or by knockout of Gm15551 in in iBAT of 12-week-old mice (LRT, *n* = 3 or 5, *p* < 0.001).

## Data Availability

The data presented in this study are openly available in GEO, reference number GSE200656.

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
