# Peer review of "Comprehensive Transcriptional Profiling and Mouse Phenotyping Reveals Dispensable Role for Adipose Tissue Selective Long Noncoding RNA Gm15551"

_ncrna, 2022, doi:10.3390/ncrna8030032_

Round 1
Reviewer 1 Report
Summary:
The cold and nutrient activated brown adipose tissue (BAT) is capable of protecting mice against diet-induced obesity and improving insulin response and glucose tolerance. Authors performed a comprehensive transcriptome profiling on BAT mice challenged with cold temperature and high fat diet as well as on seven metabolically active tissue pairs. Transcriptome profiling led to identification of multiple BAT associated lncRNAs and they selected Gm15551 for further functional studies. Gm15551 was found to be highly up regulated during adipogenesis, was found to have adipose tissue specific expression as well as down regulated upon β-adrenergic activation in mature adipocytes. Interestingly, Gm15551 was not associated with any phenotypes in vitro as well as in vivo upon its silencing or over expression.
Major Points:
All pertaining to loss of function and gain of function studies:
line 219-231: Regarding in vitro loss as well as gain of function studies, authors could consider using a positive control since this could be entirely due to technical issues. Positive control gapmers as well as a positive control for CRISPRa should validate the workings of the protocol.
Additionally in case of CRISPRa, authors could test more sgRNAs since sometimes, even though being specific, over-expression itself is not enough to elicit a biological response.
line 377-379: Can authors confirm the primer sequence was not overlapping with plasmid sequence to check for over expression ? Ideally with a good over expression, if specific, other transcriptomic changes of directly related markers should be observed. Efficiency of over expression could be misrepresented if primer sequence overlaps with the plasmic sequence.
Regarding loss of function studies: Authors could try using siRNAs since gapmers sometimes are not efficient in inducing biological response and are associated with higher toxicity to cells (could explain different effects in transcriptome signatures with different gapmers). These all could could very well just be off target effects
Minor Points:
line 110-114: Authors need to clarify if isolated total RNA was treated with DNAse before reverse transcription.
Overall, this study is well designed, manuscript is well written and investigates critical questions with a broad interest to the scientific community. However for a lncRNA to be used as a marker of any condition or a disease, effect of its loss and gain should be more clearly demonstrated. Authors could consider experimenting bit more with loss and gain of function experiments using multiple approaches to be sure of the lack of the phenotype since using two sgRNAs for GOF or two LNA gapmers is not sufficient evidence.
Reviewer 2 Report
In this manuscript, Engelhard et al. analyzed the transcriptome data of mouse in the cold and high-fat-diet stimulation conditions and picked up one lncRNA to validate its functions in BAT metabolism. Although the authors cannot pinpoint the specific function of the lncRNA, the characterization steps are valuable. The manuscript is much improved after the revision. I just have one comment on this work:
For the KD work, can authors analyze the off-target effects by searching for the potential loci and validate their expression levels? It is very uncommon that different knocking down tools have few overlapped target genes. As LNA is shorter antisense oligos, the authors must be very cautious in dealing with the potential off-target effect.
Reviewer 3 Report
The manuscript from Engelhard et al. found Gm15551 was an adipose specific lncRNA and can be up-regulated by the cold treatment by analyzing RNA-seq data. The authors also performed both in vivo and in vitro assays to study the function of Gm15551. While they have made an amount of effort for experimental verification, they didn’t observe any interesting phenotypes of gain or loss of function of Gm15551. It is of the opinion of this reviewer that the authors should discuss more why there were no effects of Gm15551.
Major comments:
1. It is interesting that transcriptional regulator Prdm16 and Pprag can bind to Gm15551 genome loci. However, why was their binding profile different? Why Pprag binds to the intron region?
2. Where is the genotyping data of ΔGm15551?
3. The author should discuss more about why Gm15551 didn’t affect any phenotypes.
4. Since Gm15551 didn’t affect any phenotypes, I am curious that is there any homologous gene of Gm15551 in the mice genome?
5. Could authors provide a table listing all the differentially expressed genes regulated by cold treatment?
6. Figure 1D showed there were other lncRNAs with even higher fold change than Gm15551. Why did authors choose Gm15551 as a potential target but not others?
7. Gm15551 contains two exons. The authors only knocked out the exon1 of Gm15551 in vivo. I am wondering whether the exon2 still can be expressed after KO exon1? Could authors provide a similar RNA expression panel like Figure 2C for ΔGm15551 RNA-seq data.
8. The author should provide more details for ChIP-seq data analysis in methods. For example, how to call the peaks? How to normalize with Input samples?
Minor comments:
1. Line 79 contained double space, please correct them.
2. The text size was different in line186.
3. “H3K327ac” and “H3K327me3” were wrong histone marker names, kindly rectify.
4. Figure2A, please provide statistical test and p value.
5. Figure 2C, please label the track name for Gm15551 and its exon number.
